# Honey-Mediated Wound Healing: H_2_O_2_ Entry through AQP3 Determines Extracellular Ca^2+^ Influx

**DOI:** 10.3390/ijms20030764

**Published:** 2019-02-11

**Authors:** Simona Martinotti, Umberto Laforenza, Mauro Patrone, Francesco Moccia, Elia Ranzato

**Affiliations:** 1DiSIT—Dipartimento di Scienze e Innovazione Tecnologica, University of Piemonte Orientale, Viale Teresa Michel 11, 15121 Alessandria, Italy; mauro.patrone@uniupo.it; 2Department of Molecular Medicine, University of Pavia, 27100 Pavia, Italy; lumberto@unipv.it; 3Department of Biology and Biotechnology “L. Spallanzani”, University of Pavia, 27100 Pavia, Italy; francesco.moccia@unipv.it; 4DiSIT—Dipartimento di Scienze e Innovazione Tecnologica, University of Piemonte Orientale, Piazza Sant’Eusebio 5, 13100 Vercelli, Italy; elia.ranzato@uniupo.it

**Keywords:** AQP3, Ca^2+^ signaling, hydrogen peroxide, honey, wound healing

## Abstract

Since Biblical times, honey has been utilized in “folk medicine”, and in recent decades the positive qualities of honey have been re-discovered and are gaining acceptance. Scientific literature states that honey has been successfully utilized on infections not responding to classic antiseptic and antibiotic therapy, because of its intrinsic H_2_O_2_ production. In our study, we demonstrated the involvement of H_2_O_2_ as a main mediator of honey regenerative effects on an immortalized human keratinocyte cell line. We observed that this extracellularly released H_2_O_2_ could pass across the plasma membrane through a specific aquaporin (i.e., AQP3). Once in the cytoplasm H_2_O_2_, in turn, induces the entry of extracellular Ca^2+^ through Melastatin Transient Receptor Potential 2 (TRPM2) and Orai1 channels. Honey-induced extracellular Ca^2+^ entry results in wound healing, which is consistent with the role played by Ca^2+^ signaling in tissue regeneration. This is the first report showing that honey exposure increases intracellular Ca^2+^ concentration ([Ca^2+^]_i_), due to H_2_O_2_ production and redox regulation of Ca^2+^-permeable ion channels, opening up a new horizon for the utilization of the honey as a beneficial tool.

## 1. Introduction

Wound healing is of countless significance for skin medicine and a precise focus is set on natural compounds.

Honey has been utilized as medicine in many cultures for a long time [1]. Honey is primarily a sugar-rich food source for bees. There are at least about 200 compounds described in honey, and it consists of water, mainly sugars, and other substances such as organic acids, proteins (enzymes), vitamins (i.e., niacin, vitamin B6, thiamine, riboflavin and pantothenic acid), minerals (including copper, calcium, etc.) [2].

In spite of a wide amount of literature about honey clinical practices, the subjacent mechanisms of action are still largely unclear. Accumulating records propose the contribution of specific physiological mechanisms, such as angiogenesis, granulation, and epithelialization [3].

We have recently elucidated the modulatory effects of different honey types in an in vitro model of keratinocytes and fibroblasts [4]. Our data have shown that honey treatments are poorly cytotoxic on skin cells, confirming that honey can be used safely not only for external applications on healthy skin, but also as a dressing on wounds. In addition, data coming from scratch wound observations as well as from cell migration assays have shown beyond any reasonable doubt that honey produces a striking increase of the wound repair abilities of keratinocytes and fibroblasts [4]. Taken together, our previous observations consistently suggest that honey is generally active in helping wound closure, thus confirming a bulk of anecdotal and scientific evidence [5,6].

The high concentrations of sugars (about 80%) combined with less than 1% of water causes osmotic stress preventing microorganism growth. Moreover, already during the 1960s, hydrogen peroxide (H_2_O_2_) was recognized as a major antibacterial compound present in honey. In fact, honey continuously provides H_2_O_2_, which is generated by glucose oxidase-mediated conversion of glucose [7].

Specific aquaporins (AQPs) facilitate the passive diffusion of H_2_O_2_ across the biological membranes [8]. AQP-mediated transmembrane transport of H_2_O_2_ is of physiological importance for further downstream signaling events, such as the onset of intracellular Ca^2+^ signals [8].

In wound healing, cellular reactions to injury are associated with an increase in intracellular Ca^2+^ concentration ([Ca^2+^]_i_) which activates multiple Ca^2+^-dependent processes associated to tissue regeneration, including proliferation, motility and differentiation [9,10,11]. Accordingly, Ca^2+^ has a main role in cellular homeostasis and is utilized by a plethora of extracellular cues to finely regulate cell fate [12,13,14]. Recent studies have demonstrated that also H_2_O_2_ may signal through an increase in [Ca^2+^]_i_ by activating Ca^2+^-permeable channels located either on the endoplasmic reticulum (ER) [15], the largest endogenous Ca^2+^ reservoir, or on the plasma membrane. Accordingly, H_2_O_2_ has been found to induce extracellular Ca^2+^ entry though Orai1 and Melastatin Transient Receptor Potential 2 (TRPM2) [16,17,18,19]. Notably, AQPs have long been known to regulate the [Ca^2+^]_i_ by modulating the osmotic gradient across the plasma membrane [20]. However, it is still unclear whether AQPs influence intracellular Ca^2+^ dynamics by mediating extracellular H_2_O_2_ entry.

Herein, we sought to investigate the functional interaction among extracellular H_2_O_2_, AQPs and plasmalemmal Ca^2+^-permeable channels in honey-induced skin regeneration. Our findings shed novel light on the intracellular mechanisms by which honey uses reactive oxygen species (ROS) to induce wound closure and hint at H_2_O_2_ as a key signaling messenger to induce tissue remodeling due to its ability to trigger intracellular Ca^2+^ signals. 

## 2. Results

### 2.1. Honey Induces Ca^2+^ Signals in a Dose-Dependent Manner

First at all, we decided to investigate if different honey types are able to induce variations in [Ca^2+^]_i_. Therefore, we assessed intracellular Ca^2+^ variations induced after honey exposure, by using time-lapse confocal microscopy imaging of HaCaT cells loaded with the fluorescent Ca^2+^ probe Fluo-3/AM. For honey concentration of 4% *v*/*v*, the calcein-AM cytotoxicity assay carried out on HaCaT cells previously demonstrated that this dose did not affect cell viability [4].

Preliminary observations showed that the [Ca^2+^]_i_, sampled at 10-s intervals, did not undergo spontaneous oscillations in control conditions (Figure 1A). The analysis of confocal imaging of Fluo-3/AM loaded cells exposed to 4% *v*/*v* honey samples revealed that manuka honey triggered a single, large [Ca^2+^]_i_ spike. The spike started a few seconds after honey exposure, reached a peak within 60–90 s, and decayed to a plateau level of intermediate amplitude in approximately 100–200 s (Figure 1A,B).

Then, we tested the effects on [Ca^2+^]_i_ after treatment with other honey samples, i.e., buckwheat and acacia honey type (Figure 1A,B). Buckwheat honey triggered a biphasic increase in [Ca^2+^]_i_ which was similar to that induced by manuka honey, but displayed a lower amplitude (Figure 1A,B). Conversely, acacia honey evoked a slow increase in [Ca^2+^]_i_ which was significantly (*p* < 0.05) reduced as compared to manuka- and acacia-induced intracellular Ca^2+^ variations (Figure 1A,B). Based on these evidences, we reasoned that manuka honey was the most suitable type of honey to investigate the role of intracellular Ca^2+^ signaling in honey-induced wound repair. We also evaluated the dose-response relationship of the Ca^2+^ response to manuka honey by examining a range of concentrations (1, 2 and 4% *v*/*v*). As shown in Figure 1C,D, 4% *v*/*v* manuka honey induced the largest increase in [Ca^2+^]_i_ in HaCaT cells and was, therefore, employed throughout the remainder of this investigation (Figure 1C,D).

### 2.2. The Ca^2+^ Response to Manuka Honey Requires Extracellular Ca^2+^ Entry and the Intracellular Production of Hydrogen Peroxide

Intracellular Ca^2+^ signals can be generated by the opening of Ca^2+^-permeable channels which are located either on the plasma membrane or are embedded within the membrane of intracellular organelles, such as the ER [12,14]. We found that the Ca^2+^ response to 4% *v*/*v* manuka honey disappeared in Ca^2+^-free medium (Figure 2A,B). Therefore, extracellular Ca^2+^ entry is the main pathway underlying-induced elevation in [Ca^2+^]_i_ in HaCaT cells. It is already known that honey samples induce H_2_O_2_ production in cell cultures, thereby causing an increase in intracellular H_2_O_2_ levels [16]. By using the xylenol orange assay, we evaluated the dose-dependent production of H_2_O_2_ induced by the different honey types (i.e., acacia, buckwheat and manuka, Figure 3A). 5% manuka honey releases in the culture medium around 50 µM H_2_O_2_. We also explored intracellular ROS levels by recording the fluorescence of DHR-123 loaded HaCaT cells in a microplate reader. As shown in Figure 3B, 4% *v*/*v* manuka honey induced a sustained rise in intracellular ROS levels. 

We then evaluated whether the addition of exogenous catalase (CAT), which scavenges H_2_O_2_, reduces manuka honey-induced Ca^2+^ signals. Confocal imaging showed that, pre-treatment of cells with different doses of CAT, reduces (500 U) or abrogates (1000 U) the Ca^2+^ response to manuka honey (Figure 3C,D). Therefore, H_2_O_2_ is the most likely candidate to trigger manuka honey-induced Ca^2+^ signaling. To further corroborate this notion, we assessed the Ca^2+^ variations induced by 50 and 100 µM H_2_O_2_. Confocal imaging showed that 50 µM H_2_O_2_ induces a biphasic increase in [Ca^2+^]_i_ similar to that induced by manuka honey, although the time to peak was slightly slower and the early phase of the plateau phase slightly higher. 100 µM H_2_O_2_ triggered a progressive and sustained rise of [Ca^2+^]_i_, reaching values that were remarkably higher than those achieved by manuka honey and not compatible with cell survival (Figure 3E,F).

### 2.3. Role of Specific Inhibitors of Ca^2+^ Entry

To disclose the mechanism(s) responsible for of the Ca^2+^ response to manuka honey, we focused on the role of TRPM2 and Orai1. Accordingly, H_2_O_2_ has long been known to activate TRPM2 [17,18]; in addition, recent studies revealed that H_2_O_2_ may also activate Orai1-dependent Ca^2+^ entry [21,22]. Confocal imaging showed that HaCaT cell preincubation with econazole, a TRPM2 inhibitor [23,24], was able to reduce the [Ca^2+^]_i_ increase caused by manuka exposure. Econazole (10 µM, 30 min) significantly (*p* < 0.05) reduced the amplitude of both the peak and the plateau phase of the Ca^2+^ response to manuka honey (Figure 4A,B). As econazole is not selective to TRPM2 channels, we down-regulated its expression with a specific RNAi selectively targeting TRPM2 (Figure 4D), and observed an almost complete abrogation of the Ca^2+^ response to manuka honey exposure (Figure 4E,F).

Likewise, Pyr6 (1 µM, 30 min), a selective inhibitor of Orai1 channels, caused a significant decrease of both phases of manuka honey-induced elevation in [Ca^2+^]_i_ (Figure 4A,B). As Pyr6 selectively targets Orai1, but not TRP channels [11,22,25], we did not silence Orai1. These data strongly suggest that TRPM2 and Orai1 mediate manuka honey-induced extracellular Ca^2+^ entry. In agreement with this hypothesis, when HaCaT cells were pretreated with both econazole and Pyr6, the Ca^2+^ response to manuka honey was fully abrogated. Cell incubation with vehicle DMSO, used alone, was ineffective on intracellular Ca^2+^ dynamics (not shown). 

In order to extend at molecular levels the findings obtained during Ca^2+^ imaging recordings, we used qPCR to confirm that Stim1-2, Orai1-3 and TRPM2 transcripts are expressed in HaCaT cells (Figure 4C).

### 2.4. Role of AQP3 in H_2_O_2_ Influx

As discussed in Section 2.2, the Ca^2+^ response to manuka honey requires an increase in intracellular ROS levels which follows H_2_O_2_ liberation in the culture medium. Therefore, we hypothesized that, once released in the extracellular milieu, H_2_O_2_ diffuses across the plasma membrane to increase intracellular ROS concentration and activate external Ca^2+^ entry. AQPs are integral membrane proteins acting as channels in the transfer of water across the membrane, playing an important role in skin hydration [26,27,28]. In addition, AQP3 is able to mediate the transport of H_2_O_2_ in different tissues and conditions [29,30,31]. Therefore, we first decided to quantify the basal expression of some AQPs and to assess the variation in their expression levels after honey exposure, by means of qPCR analysis and immunoblotting. In Figure 5A,B, we analyzed the basal expression of AQP-1, -3, -4, -5, -6, -8 and -9 transcripts in keratinocytes, but only the expression of AQP3 was remarkable enhanced after honey exposure, in particular after treatment with manuka honey. Immunoblotting results confirmed the AQP3 upregulation after manuka treatment. Therefore, we decided to investigate the role of AQP3 as a mediator of H_2_O_2_ signaling in the Ca^2+^ response to manuka honey by using a specific RNAi. As expected, silencing of AQP3 prevented the increase in intracellular ROS despite manuka honey exposure (Figure 5G), thereby completely abrogating the rapid increase in [Ca^2+^]_i_ (Figure 5C,D).

To assess the role played by AQP3 in manuka honey-induced wound repair, we performed the scratch wound assay after silencing of AQP3 by RNAi (as shown in Figure 5F). In silenced keratinocytes (see Figure 5E), despite honey treatment, we could not observe any significant variation in the rate of wound closure, confirming the pivotal role of AQP3 in mediating wound closure.

To disclose the functional role of manuka honey-induced Ca^2+^ signals in tissue regeneration, we performed the scratch wound assay in presence or not of econazole and Pyr6 (Figure 6A,C). We observed that these inhibitors significantly reduced the wound healing rate, thereby confirming the role of TRPM2 and Orai1 in manuka honey-dependent tissue regeneration. Moreover, also CAT significantly (*p* < 0.05) reduced manuka honey-induced wound repair, in agreement with the role played by H_2_O_2_ in TRPM2 and ORA1 activation (Figure 6B,C).

## 3. Discussion

The essential role of intracellular Ca^2+^ in wound repair and regeneration processes has been suggested by in vitro studies on scratch wounded layers of fibroblasts and endothelial cells [10,32].

Even more striking evidence has come from an in vitro study on HaCaT keratinocytes, showing that the cell-permeant Ca^2+^ chelator BAPTA inhibits both the wound closure and chemoattraction effects [33].

Wound-induced intracellular Ca^2+^ waves have been also observed in endothelial cells [34], whereas [Ca^2+^]_i_ rises have been found to promote cell growth and movement in endothelial repair [35]. Finally, in vivo evidence of the regulatory role of [Ca^2+^]_i_ in wound healing comes from the finding of high calmodulin levels in mitotic basal keratinocytes during the proliferative phase of rat wounds [36].

However, until now, no data are available about component(s) (other than antibacterial) of the honey able to induce/accelerate wound healing. Honey samples are able to release, also in in vitro conditions, H_2_O_2_ [37], and this H_2_O_2_ can contribute to their antibacterial properties [38].

In our study, we demonstrated the involvement of H_2_O_2_ as a main mediator of honey regenerative effects on HaCaT cells, a spontaneously immortalized human keratinocyte line 

We confirmed that this extracellularly released H_2_O_2_ could pass across the plasma membrane through a specific aquaporin (i.e., AQP3). AQP3 has been already demonstrated to play a pivotal role in hydration of mammalian skin as well as in the regulation of proliferation and differentiation of keratinocytes [27]. Once in the cytoplasm, H_2_O_2_ induces the entry of extracellular Ca^2+^ through TRPM2 and Orai1 channels by a ROS-dependent mechanism [39]. The following pieces of evidence support this model. First, manuka honey failed to elicit a detectable increase in [Ca^2+^]_i_ in the absence of extracellular Ca^2+^. Second, the Ca^2+^ response to manuka honey was mimicked by 50 µM H_2_O_2_ and inhibited in a dose-dependent manner by CAT, an H_2_O_2_ scavenger. Accordingly, honey samples may release H_2_O_2_ that is produced by the enzyme glucose oxidase and is responsible for the antiseptic effect of honey dressing [40]. Despite the production of H_2_O_2_ by honey, a harmful oxidizing effect is not observed on skin cells due to the honey polyphenolic component, which can antagonize this pro-oxidant action. In agreement with this hypothesis, the use of generic fluorescent H_2_O_2_ indicators revealed local sites of H_2_O_2_ production within the cytoplasm, including ER, mitochondria and plasma membrane [41,42,43]. This mechanism could also explain the different kinetics and amplitude of the Ca^2+^ response induced by the three different types of honey we probed. Accordingly, the H_2_O_2_-releasing ability is manuka > buckwheat >> acacia. Third, genetic silencing of AQP3 with a selective RNAi prevented the increase in intracellular ROS levels and abrogated the Ca^2+^ response to manuka honey. Accordingly, AQP3 has recently been shown to mediate H_2_O_2_ uptake and accumulation within the cytosolic leaflet of the plasma membrane, thereby generating a signaling nanodomain that could gate adjacent H_2_O_2_-sensitive channels [41]. Conversely, AQP3 is not permeable to extracellular Ca^2+^, while there is not report about the recruitment of an intracellular signaling pathway leading to H_2_O_2_ production upon honey exposure. These data, therefore, strongly suggest that AQP3 mediates H_2_O_2_ entry into the cytosplasm. Fourth, H_2_O_2_ may activate both TRPM2- and Orai1-mediated extracellular Ca^2+^ entry by acting from the cytosolic side. As to TRPM2, it is still unclear whether H_2_O_2_ directly gates TRPM2 [44] or does so by inducing ADP ribose (ADPr) mobilization from mitochondria [45,46]. H_2_O_2_ could then facilitate ADPr-dependent TRPM2 activation [45]. Orai1, in turn, mediates the so-called store-operated Ca^2+^ entry (SOCE), which is activated upon depletion of the ER Ca^2+^ store [25]. The drop in ER Ca^2+^ levels is sensed by STIM1, an ER-resident single-pass protein which oligomerizes and redistributes towards ER-plasma membrane junctions, known as puncta, thereby binding to and gating ORAI1 [25]. In agreement with our observations, earlier studies revealed that H_2_O_2_ may activate Orai1 by promoting Stim1 oligomerization and relocalization from perinuclear ER to plasma membrane puncta through S-glutathionylation of cysteine 56 in COS7 cells [47]. Likewise, H_2_O_2_ was found to stimulate Orai1 in a Stim1-dependent manner in HEK293 and DT40 cells [21] and in coronary vascular smooth muscle cells [22]. 

Honey-induced Ca^2+^ entry through TRPM2 and Orai1 channels, in turn, accelerated wound closure in human keratinocytes. We have recently demonstrated that honey boosted the rate of wound repair in HaCaT cells by promoting migration rather than proliferation [48]. Of note, we found that honey-induced wound closure was inhibited by BAPTA, a membrane permeable buffer of intracellular Ca^2+^ levels [4,10,49], and PD98059, which selectively blocks the extracellular signal-regulated kinase (ERK) pathway [5,50]. Herein, we demonstrated that honey-induced wound healing was severely inhibited by blocking extracellular Ca^2+^ entry by the genetic suppression of AQP3 (which prevents H_2_O_2_ entry), H_2_O_2_ scavenging (with CAT) and the pharmacological blockade of TRPM2 and Orai1 channels (with econazole and Pyr6, respectively). Therefore, TRPM2 and Orai1 provide the major sources of Ca^2+^ to recruit the Ca^2+^-dependent machinery driving HaCaT cell migration. Other works have demonstrated that intracellular Ca^2+^ signaling may induce cell migration by engaging the ERK pathway [10,50,51,52,53]. For instance, ORAI1-mediated Ca^2+^ entry stimulated human melanoma cell migration by recruiting Ca^2+^/Calmodulin-dependent kinase II (CaMKII) to induce Raf-1 phosphorylation, thereby activating the ERK pathway [53]. Likewise, TRPM2 has been shown to drive migration in several cell types [54,55], although the downstream signaling machinery remains to be elucidated.

In conclusion, we demonstrated that the presence of H_2_O_2_ in the extracellular space (due to honey’s glucose oxidase enzymatic activity [7]) causes a change in the intracellular levels of Ca^2+^.

The present results can be explained by the following model (see Figure 7):Honey produces H_2_O_2_ in the extracellular spaceAQP3 mediates the H_2_O_2_ entry in to the cytosol [41]H_2_O_2_ activates TRPM2 and Orai1 determining Ca^2+^ entry [56] from the outside that induces wound closure.

This is the first report showing that honey exposure affects [Ca^2+^]_i_ regulation, due to H_2_O_2_ production and redox regulation of ion channels. 

These data lay the foundations for scientific and rational use of honey as a wound repair agent, with the understanding of cellular and molecular mechanisms underlying the effects on skin cells. 

Moreover, our observations open up a new horizon for the use of the honey as a beneficial tool in the management of skin disorder by the modulation of aquaporin expression.

## 4. Materials and Methods

### 4.1. Honey Samples

Honey samples were obtained from Yamada Apiculture Center, Inc., Tomata-Gun, Okayama (Japan).

### 4.2. Cell Culture and Reagents

All reagents were from Sigma-Aldrich (St. Louis, MO, USA), unless otherwise indicated. 

HaCaT cells are immortalized human skin keratinocytes that mimic many properties of normal epidermal keratinocytes, are not invasive, and can differentiate under appropriate experimental conditions [33]. 

Cells were maintained at 37 °C, 5% CO_2_, in DMEM supplemented with 10% foetal bovine serum (FBS, Euroclone, Milan, Italy) and 1% antibiotic mixture.

### 4.3. Measurements of Free Cytosolic Ca^2+^ Concentration ([Ca^2+^]_i_)

HaCaT cells were plated on glass-base dishes (Iwaki Glass, Inc., Tokyo, Japan). They were allowed to settle overnight, and then loaded with the cell-permeant, fluorescent calcium probe Fluo-3/AM (20 mM) in the dark at 37 °C for 30 min. The loading buffer is composed of (mM) 10 HEPES pH 7.4, 10 glucose, 140 NaCl, 2 CaCl_2_, 1 MgCl_2_, 5 KCl. For Ca^2+^-free experiments, this ion was omitted from the loading buffer [57,58].

After probe loading and washing, cells were analyzed through confocal time-lapse analysis, using a Zeiss LSM 510 confocal system interfaced with a Zeiss Axiovert 100 M microscope (Carl Zeiss Inc., Oberkochen, Germany). 

Excitation was obtained by the 488 nm line of an Ar laser, and emission was collected using a 505–550 bandpass filter. The laser power was reduced to 15% in order to lower probe bleaching. 

Confocal imaging was performed with a resolution of 512 × 512 pixels at 256 intensity values, with a framing rate of 1 frame/5 s. 

Several cells were viewed together through a 20× Plan-Neofluar Zeiss objective (0.5 NA). Fluo-3/AM fluorescence was analyzed in digitized images as the average value over defined contours of individual cells, using the ROI-mean tool of the Zeiss LSM 510 2.01 software (Carl Zeiss Microscopy, Feldbach, Switzerland). Fluo-3 calibration was achieved by the following Equation (1) [59].
Ca^2+^ = *K*_d_ × (*F* − *F*min)/(*F*max − *F*)(1)
where *K*_d_ = 400 nmol/L. *F*max and *F*min are maximum and minimum fluorescence intensities obtained by Fluo-3/AM calibration after cell exposure to 500 μM A23187 for about 10 min, followed by addition of 20 mM EDTA.

### 4.4. H_2_O_2_ Assays

The formation of H_2_O_2_ induced by manuka in culture medium was determined by xylenol orange, a colorimetric assay, as previously described [58,60].

### 4.5. Measurement of Intracellular ROS

Intracellular levels of ROS were assessed using the fluorescent dye precursor dihydrorhodamine (DHR) 123, which is converted to fluorescent rhodamine 123 upon reaction with ROS. Cells were seeded in 96-well plates, allowed to settle overnight, and loaded for 30 min at room temperature in the dark with DHR-123 (30 µM) in a loading buffer consisting of (mM) 10 glucose, 10 Hepes, 140 NaCl, 2 CaCl_2_, 1 MgCl_2_, and 5 KCl, pH 7.4. Cells were then washed with loading buffer and fluorescence collected in a fluorescence microplate reader, by using 485-nm excitation and 530-nm emission filters. Data of ROS production were expressed as fluorescence arbitrary units [58].

### 4.6. Quantitative Reverse Transcriptase PCR (qRT-PCR) and RNA Interference (siRNA)

Cells were settled in multi-well plates for 24 h and then subjected to the indicated experimental conditions. NucleoSpin RNAII Kit (Macherey-Nagel, Düren, Germany) was then utilized to extract total RNA. Complementary DNA was synthesized from RNA using the Transcriptor First Strand cDNA Synthesis Kit (Roche Diagnostics GmbH, Penzberg, Germany). 

qRT-PCR was performed using Power Sybr Green Mastermix (Ambion Austin, TX, USA) and KiCqStart^®^ SYBR^®^ Green Primers (Sigma-Aldrich, Table 1) in a CFX384 Real-Time PCR Detection System (Bio-Rad Laboratories, Hercules, CA, USA). Gene expression was measured by the ∆∆Ct method.

For AQP3, RNAi was performed by transfecting cells with 5 µM siRNA oligonucleotides (Sigma-Aldrich, Table 2) or with equimolar scramble siRNA by using the N-ter Nanoparticle siRNA Transfection System (Sigma-Aldrich). 

For TRPM2, RNAi was performed using esiRNA (esiRNA Human TRPM2, cat. no. EHU133821, Sigma-Aldrich).

Scramble siRNA was obtained using commercial non-targeting siRNA (MISSION siRNA Universal Negative Control). 

Cells were collected at 24 h after transfection and used for the experiments.

### 4.7. Immunoblotting

Cells were lysed in RIPA buffer (supplemented with a protease and phosphatase inhibitor cocktail) and homogenates were solubilized in Laemmli buffer [61]. 15–30 μg solubilized proteins were subjected to 12.5% SDS-polyacrilamide gel electrophoresis and transferred to the Hybond-P PVDF Membrane (GE Healthcare, Chicago, IL, USA) by electroelution. The membranes were incubated overnight with anti-AQP3 antibody produced in rabbit (SAB5200111; Sigma, Italy) diluted 1:1000 in TBS and 0.1% Tween-20. The membranes were washed and incubated for 1 h with goat anti-rabbit IgG antibody, peroxidase conjugated (AP132P; Millipore, Burlington, MA, USA) diluted 1:100,000 in blocking solution. The bands were detected with ECL™ Select western blotting detection system (GE Healthcare). Pre-stained molecular weight markers (ab116028, Abcam) were used to estimate the molecular weight of the bands. Blots were stripped and re-probed with RabMAb anti β-2-microglobulin antibody ([EP2978Y] ab75853; Abcam, 1:10000) or with anti β-actin polyclonal antibody (Cat.n. AB-81599; Immunological sciences, USA, 1:2000) as housekeeping.

### 4.8. Scratch Wound Test

Scratch wounds were made in confluent layers HaCaT cells by using a sterile 0.1–10 µL pipette tip. After washing away suspended cells, cultures were refed with complete medium in the presence of honey for 24 h. After cell exposures, cells were fixed in 3.7% formaldehyde in PBS for 30 min, stained for 30 min at room temperature with 0.1% toluidine blue. The width of the wound space was assessed at wounding and at the end of treatments, using an inverted microscope (Leica Microsystems, Wetzlar, Germany) equipped with a digital camera. Digitized pictures of wounds were analyzed using the NIH Image J software. In a typical experiment, each group consisted of three different plates, i.e., a total of six wounds. Four measurements of wound width were made for each wound at randomly chosen points. Wound closure was measured as the difference between wound width at 0 h and at 24 h. Measurements were made by a single observer unaware of the treatments. Wound closure rates were determined as the difference between wound width at 0 and 24 h. The value of the control was set at 100 percent in order to obtain wound closure rate %.

### 4.9. Statistical Analysis

Data were analyzed by using the Instat software package (GraphPad Software, Inc, San Diego, CA, USA). 

## Figures and Tables

**Figure 1 ijms-20-00764-f001:**
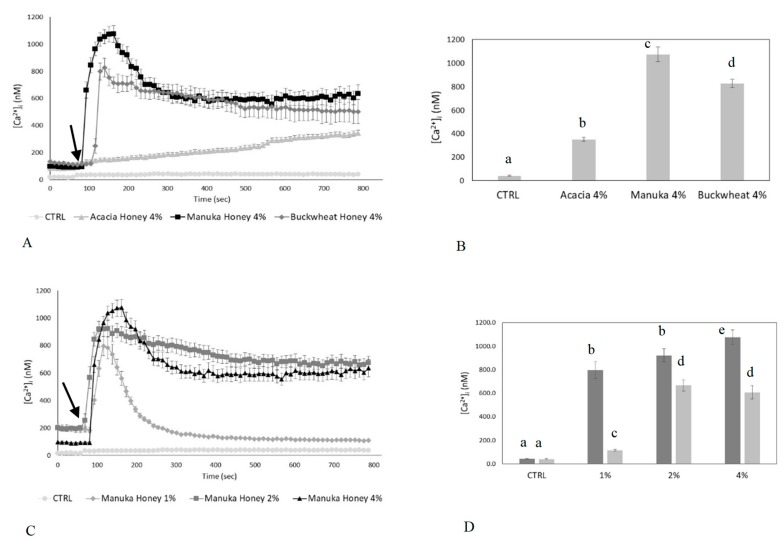
Honey induces an increase in intracellular Ca^2+^ concentration in human keratinocytes. (**A**) [Ca^2+^]_i_ variations recorded at 10-s intervals, showing no variations in control conditions, and distinct patterns of Ca^2+^ signaling after exposure to different Honey samples (i.e., Acacia, Manuka, Buckwheat 4% *v*/*v*). The arrow indicates the addition of different honey types after 60 s. Data are presented as means ± SEM. of [Ca^2+^]_i_ traces recorded in different cells. Number of cells: CTRL: 20 cells from 2 experiments (exp); acacia honey: 40 cells from 3 exp; manuka honey: 30 cells from 3 exp; buckwheat honey: 40 cells from 3 exp. (**B**) Mean ± SEM of the peak Ca^2+^ response induced by treatment with 4% *v*/*v* of different types of honey. Number of cells as in **A**. Different letters above bars indicate statistical differences determined by One-way ANOVA followed by Dunnet post-test (*p* < 0.01). (**C**) Dose-response relationship of the increase in [Ca^2+^]_i_ induced by different concentration (%) *v*/*v* of manuka honey and recorded at 10-s intervals. The arrow indicates the addition of different concentrations of manuka honey after 60 s. Data are means ± SEM of [Ca^2+^]_i_ traces recorded in different cells. Number of cells: CTRL: 20 cells from 2 exp; 1% *v*/*v* manuka honey: 40 cells from 3 exp; 2% *v*/*v* manuka honey: 30 cells from 3 exp; 4% *v*/*v* manuka honey: 40 cells from 3 exp. (**D**) Mean ± SEM of the Ca^2+^ response recorded at the peak (light bars) and at the plateau (dark bars) in the presence of different concentration (%) *v*/*v* of manuka money. Number of cells as in (**C**). Different letters above the bars indicate statistical difference determined by two-way ANOVA followed by Bonferroni’s correction (*p* < 0.01).

**Figure 2 ijms-20-00764-f002:**
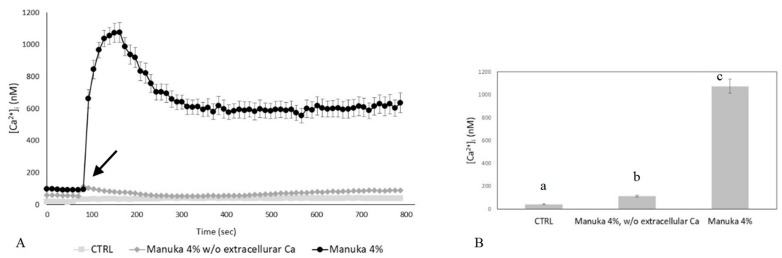
The Ca^2+^ response to manuka honey requires extracellular Ca^2+^ entry. (**A**) The Ca^2+^ response to 4% *v*/*v* manuka honey was abolished in Ca^2+^-free medium. Control cells, which were not exposed to the treatment, did not show any change in [Ca^2+^]_i_. The arrow indicates the addition of manuka honey after 60 s. Data are means ± SEM of [Ca^2+^]_i_ traces recorded in different cells. Number of cells: CTRL: 20 cells from 2 exp; manuka honey: 40 cells from 3 exp; manuka honey w/o external Ca^2+^: 30 cells from 3 exp. (**B**) Mean ± SEM of the peak Ca^2+^ response recorded under the designated treatments. Number of cells as in **A**. Different letters above bars indicate statistical difference determined by One-way ANOVA followed by Dunnet post-test (*p* < 0.01).

**Figure 3 ijms-20-00764-f003:**
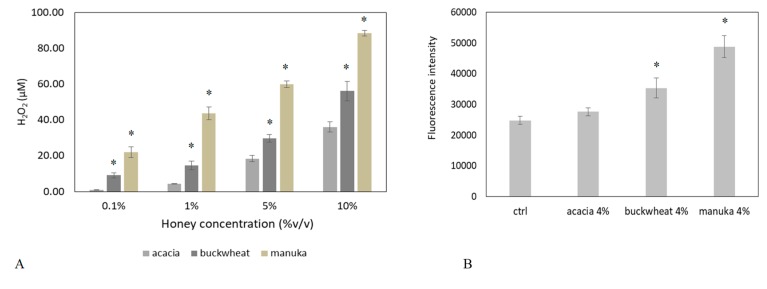
The Ca^2+^ response to manuka honey requires the intracellular production of hydrogen peroxide. (**A**) Production of H_2_O_2_ after 10 min incubation with increasing concentrations of distinct honey types. Different asterisks above bars indicate statistical difference determined One-way ANOVA followed by Bonferroni post-test (*p* < 0.01); (**B**) Fluorescence values recorded at 10 min in control cells (ctrl), or in cells incubated with different honey samples (i.e., Acacia, Manuka, Buckwheat 4% *v*/*v*) (* *p* < 0.001, Tukey’s test). Data are presented as means ± SD of rhodamine 123 fluorescence expressed in arbitrary units; *n* = 16 microplate wells from two different experiments. (**C**) The Ca^2+^ response to 4% *v*/*v* manuka honey is inhibited by Catalase (CAT, 30 min preincubation) in a dose-dependent manner. Data are presented as means ± SEM of [Ca^2+^]_i_ traces recorded in different cells. The arrow indicates the addition of manuka honey after 60 s. Number of cells: manuka honey: 40 cells from 3 exp; manuka honey + 500 U CAT: 40 cells from 3 exp; manuka honey + 1000 U CAT: 40 cells from 3 exp. (**D**) Mean ± SEM of the peak Ca^2+^ response recorded under the designated treatments. Number of cells as in **A**. Different letters above the bars indicate statistical difference determined by One-way ANOVA followed by Bonferroni post-test (*p* < 0.01). (**E**) [Ca^2+^]_i_ variations induced by 4% *v*/*v* manuka honey and by 50 and 100 µM H_2_O_2_. Data are presented as means ± SEM of [Ca^2+^]_i_ traces recorded in different cells. The arrow indicates the addition of different treatments after 60 s. Number of cells: manuka honey: 40 cells from 3 exp; 50 µM H_2_O_2_: 30 cells from 3 exp; 100 µM H_2_O_2_: 30 cells from 3 exp. (**F**) Measurements of peak Ca^2+^ response upon treatment with 4% *v*/*v* manuka honey or different H_2_O_2_ concentrations. Mean ± SEM of the peak Ca^2+^ response recorded under the designated treatments. Number of cells as in (**E**). Different letters above the bars indicate statistical difference determined by One-way ANOVA followed by Dunnet post-test (*p* < 0.01).

**Figure 4 ijms-20-00764-f004:**
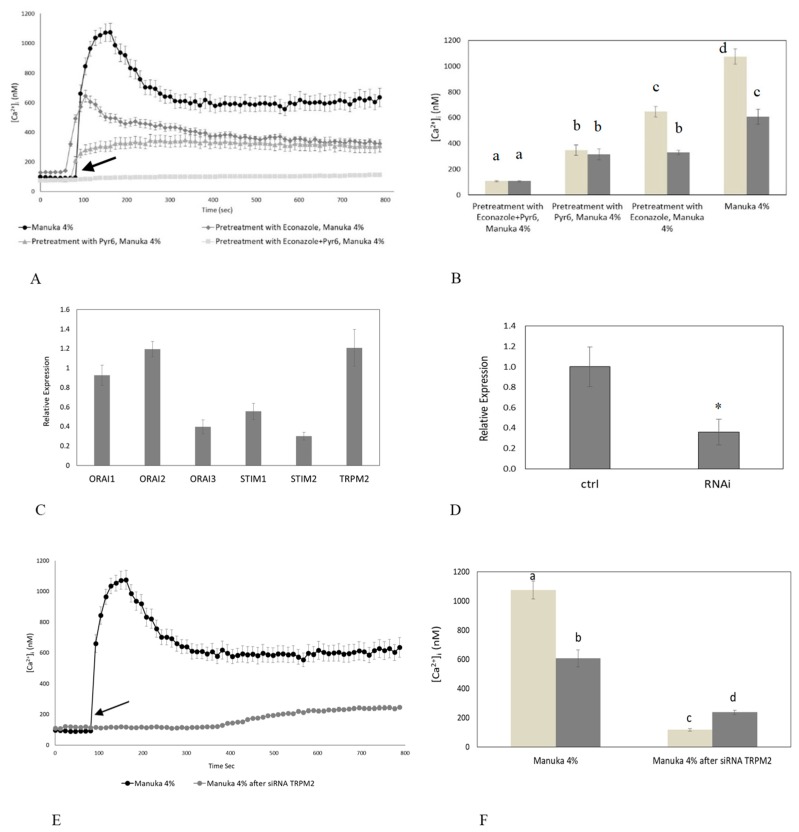
The inhibition of TRPM2 and Orai1 channels abolishes manuka honey-induced Ca^2+^ entry. (**A**) Manuka honey-induced Ca^2+^ entry was dramatically reduced in the presence of either 10 µM Econazole or 1 µM Pyr6 (30 min preincubation for each drug) and completely abolished by the simultaneous treatment with both drugs. Manuka honey was added at 4% *v*/*v*. The arrow indicates the addition of manuka honey after 60 s. Data are presented as means ± SEM of [Ca^2+^]_i_ traces recorded in different cells. Number of cells: manuka honey: 40 cells from 3 exp; manuka honey + Econazole 10 µM: 50 cells from 3 exp; manuka honey + Pyr6 1 µM: 50 cells from 3 exp; manuka honey + Econazole 10 µM + Pyr6 1 µM: 50 cells from 3 exp. (**B**) Mean ± SEM of the Ca^2+^ response to 4% *v*/*v* manuka honey recorded at the peak (light bars) and at the plateau (dark bars) under the designated treatments. Data are presented as means ± SEM of [Ca^2+^]_i_ measured by confocal imaging at peak maxima. Number of cells as in (**A**). Different letters above the bars indicate statistical difference determined by Two-way ANOVA with Bonferroni’s correction (*p* < 0.01). (**C**) The mRNA quantity of Stim1-2, Orai1-3 and TRPM2 transcripts was determined by qRT-PCR and is represented as mean relative expression ± SD (*n* = 3). (**D**) Expression of TRPM2 gene in HaCaT cells after RNAi. The mRNA quantity of TRPM2 was determined by qRT-PCR and is represented as mean relative expression ± SD (*n* = 3, * *p* < 0.001, *t*-test). (**E**) The Ca^2+^ response to 4% *v*/*v* manuka honey was suppressed in HaCaT cells transfected with a RNAi selectively targeting TRPM2. The arrow indicates the addition of manuka honey after 60 s. Data are presented as means ± SEM of [Ca^2+^]_i_ traces recorded in different cells. Number of cells: manuka honey: 40 cells from 3 exp; manuka honey after RNAi for TRPM2: 40 cells from 3 exp. (**F**). Mean ± SEM of the Ca^2+^ response to 4% *v*/*v* manuka honey recorded at the peak (light bars) and at the plateau (dark bars) under the designate treatments. Number of cells as in (**E**). Different letters above the bars indicate statistical difference determined by two-way ANOVA with Bonferroni’s correction (*p* < 0.01).

**Figure 5 ijms-20-00764-f005:**
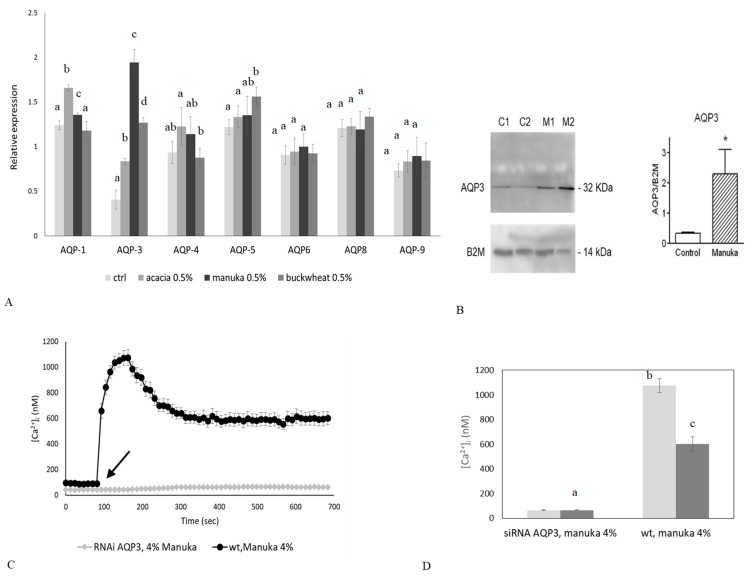
The genetic silencing of AQP3 abolishes the Ca^2+^ response to manuka honey and prevents wound closure in vitro. (**A**) Expression of AQPs genes in HaCaT cells exposed to different honey sample. The mRNA quantity of AQPs was determined by qRT-PCR and is represented as mean relative expression ± SD (*n* = 3). Different letters above bars indicate statistical difference determined by One-way ANOVA followed by Bonferroni post-test (*p* < 0.01). (**B**) Aquaporin-3 (AQP3) protein expression in HaCaT cells after manuka honey exposure. C is control condition, M is manuka honey treatment. Blots representative of three were shown. Lanes were loaded with 30 μg of proteins, probed with anti-AQP3 rabbit polyclonal antibodies and processed as described in Materials and Methods. The same blots were stripped and re-probed with anti-β-2-microglobulin (B2M) polyclonal antibody, as housekeeping. A major band of about 32 kDa was observed (* *p* < 0.001, *t*-test). (**C**) The Ca^2+^ response to 4% *v*/*v* manuka honey was suppressed in HaCaT cells transfected with a RNAi selectively targeting AQP3. The arrow indicates the addition of manuka honey after 60 s. Data are presented as means ± SEM of [Ca^2+^]_i_ traces recorded in different cells. Number of cells: manuka honey: 40 cells from 3 exp; manuka honey after RNAi for AQP3: 40 cells from 3 exp. (**D**) Mean ± SEM of the Ca^2+^ response to 4% *v*/*v* manuka honey recorded at the peak (light bars) and at the plateau (dark bars) under the designate treatments. Number of cells as in **C**. Different letters above bars indicate statistical difference determined by One-way ANOVA followed by Bonferroni post-test (*p* < 0.01). (**E**) Effect of AQP3 RNAi on honey-induced scratch wound repair of HaCaT monolayers. Wound closure rate is expressed as the difference between wound width at 0 and 24 h. Data were recorded 24 h after scratch wound healing of cells exposed to 0.1% manuka honey. Bars represent mean ± SEM of wound closure inhibition deriving from two independent experiments, each with *n* = 20. Different letters on bars indicate significant differences according to Tukey’s test (*p* < 0.01). (**F**) Aquaporin-3 (AQP3) protein expression in HaCaT cells after AQP3 RNAi. Blots representative of three were shown. Lanes were loaded with 15 μg of proteins, probed with anti-AQP3 rabbit polyclonal antibodies and processed as described in Materials and Methods. The same blots were stripped and re-probed with anti-β-actin polyclonal antibody, as housekeeping (* *p* < 0.001, *t*-test). (**G**) Fluorescence values recorded at 10 min. in scrambled or silenced (RNAi AQP3) cells, in control conditions (dark bars) or incubated with 4% manuka honey (light bars). Data are presented as means ± SD of rhodamine 123 fluorescence expressed in arbitrary units; *n* = 16 microplate wells from two different experiments. Different letters on bars indicate significant differences according to Tukey’s test (*p* < 0.01).

**Figure 6 ijms-20-00764-f006:**
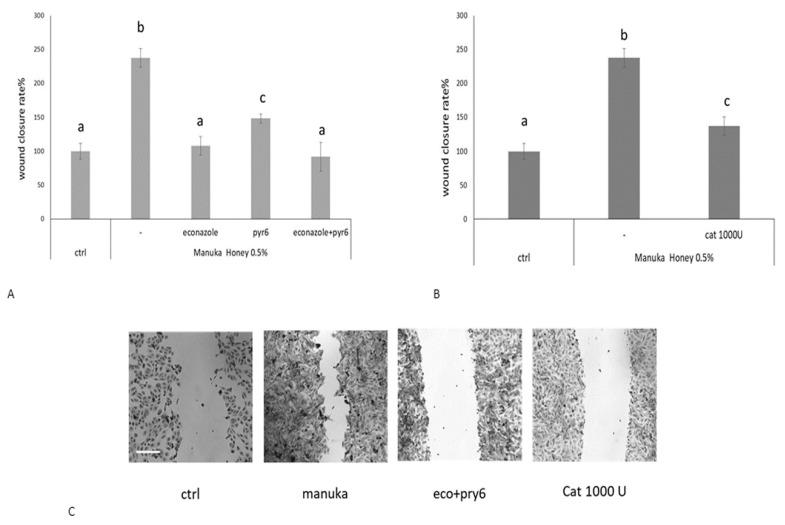
Manuka honey-induced wound closure requires hydrogen peroxide and extracellular Ca^2+^ entry. (**A**) Effect of different inhibitors on honey-induced scratch wound repair of HaCaT monolayers. Each inhibitor was used according to the procedures, which were previously shown to inhibit manuka honey-induced extracellular Ca^2+^ entry: 10 μM Econazole, 1 μM Pyr6. We also used (**B**) 1000 U CAT. Wound closure rate is expressed as the difference between wound width at 0 and 24 h. Data were recorded 24 h after scratch wound healing of cells exposed to 0.5% manuka honey, in the presence or absence of various inhibitors. Bars represent mean ± SEM of wound closure inhibition deriving from two independent experiments, each with *n* = 20. For each inhibitor, different letters above bars indicate significant differences according to Tukey’s test (*p* < 0.01). (**C**) Micrographs of scratch wounded HaCaT monolayers incubated under control conditions or in the presence of manuka honey and inhibitors and then stained with blue toluidine and observed 24 h after wounding. Scale bar, 200 μm.

**Figure 7 ijms-20-00764-f007:**
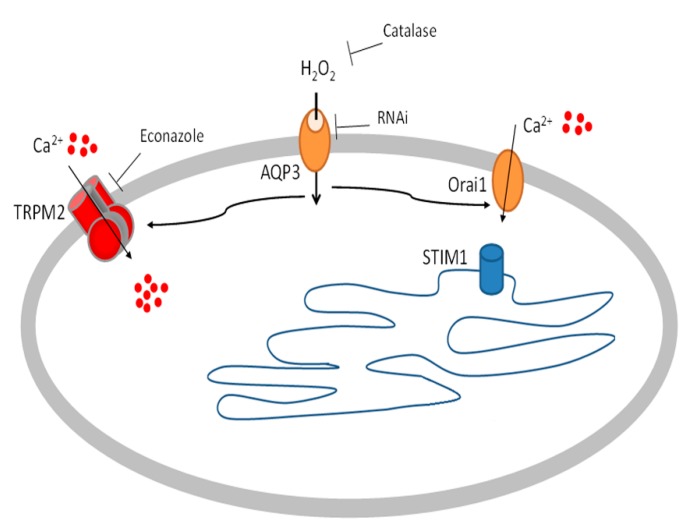
Diagram depicting the mechanism of action of manuka honey on HaCaT keratinocytes, as characterized in the study. Data showed that honey could trigger intracellular Ca^2+^ changes. The levels and profile were dependent on the source of honey and the source of the Ca^2+^ is extracellular. Honey induced the production of micromolar levels of H_2_O_2_. Catalase blocks the Ca^2+^ response indicating that it is a function of H_2_O_2_ production. Pharmacological and genetic (siRNA) inhibition of TRPM2 inhibited the Ca^2+^ response to honey. Pharmaceutical inhibition of Orai1 also diminished the Ca^2+^ response to honey. Inhibition of both types of channels completely abrogated the Ca^2+^ response. Examination of aquaporins expression found that AQP3 was upregulated upon honey exposure. siRNA for AQP3 blocked the Ca^2+^ increase elicited by honey treatment.

**Table 1 ijms-20-00764-t001:** Sequences of primers used for qRT-PCR.

Target Gene	Forward Sequence	Reverse Sequence
β-actin	5′-TCCCTGGAGAAGAGCTACGA-3′	5′-AGCACTGTGTTGGCGTACAG-3′
GADPH	5′-AATCCCATCACCATCTTCCA-3′	5′-TGGACTCCACGACGTACTCA-3′
AQP1	5′-TAAGGAGAGGAAAGTTCCAG-3′	5′-AAAGGCAGACATACACATAC-3′
AQP3	5′-CTGTGTATGTGTATGTCTGC-3′	5′-TTATGACCTGACTTCACTCC-3′
AQP4	5′-GCTGTGATTCCAAACGGACTGATC-3′	5′-CTGACTCCTGTTGTCCTCCACCTC-3′
AQP5	5′-GCTGGCACTCTGCATCTTCGC-3′	5′-AGGTAGAAGTAAAGGATGGCAGC-3′
AQP6	5′-ATTGGGATCCACTTCACTG-3′	5′-AGTGGACTGTGAACTTCC-3′
AQP8	5′-GGAGATAAGAGTATCTTGCAC-3′	5′-CTTGTCATTGCCAAATTCAG-3′
AQP9	5′-GTATTGGTAGAAACAGGAGTC-3′	5′-GGACAATCAAGATGAACGTG-3′
ORAI1	5′-ATAAGCATTTCCTGTTCTTCC-3′	5′-ACACATGTACACACTCAATG-3′
ORAI2	5′-ATTCGTATAAATGACCTGCC-3′	5′-GTGGTGGTTAGAGGTGAC-3′
ORAI3	5′-CTATCTTTGGAGGTTCAAGC-3′	5′-AGAACAAGTTTGGTGCATAG-3′
STIM1	5′-AGTGAGGATGAGAAACTCAG-3′	5′-GAACTCATCACTTTCTTCCAC-3′
STIM2	5′-GCTATTGCTAAAGATGAGGC-3′	5′-TCCAGAATTTTGTGGTCTAC-3′
TRPM2	5′-CTACCTGAAGGAGAACTACC-3′	5′-CAACCTTATTGCTGATGTCC-3′

**Table 2 ijms-20-00764-t002:** Sequences of siRNA oligonucleotides.

Target Protein	Forward Sequence	Reverse Sequence
AQP3	5′-GAGCAGAUCUGAGUGGGCA-3′	5′-UGCCCACUCAGAUCUGCUC-3′

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
