# Peer review of "Honey-Mediated Wound Healing: H_2_O_2_ Entry through AQP3 Determines Extracellular Ca^2+^ Influx"

_ijms, 2019, doi:10.3390/ijms20030764_

Round 1
Reviewer 1 Report
Purpose: to investigate possible interactions between H2O2 , Aquaporins, and Ca+2 channels.
Background: Honey (largely through osmotic stress) is antimicrobial- no discussion why it does not have the same effect on host cells. Honey also produces H2O2 which, may be antimicrobial or have an effect on gene expression. Aquaporins are known to facilitate passive diffusion of H2O2. Aquaporins can also influence Ca2 levels. Ca+2 to regulate several processes involved in wound healing.
Results:
· Showed that honey could trigger Ca changes. The levels and profile was dependent on the source of honey and the source of the calcium is extracellular
· Honey induced micromolar levels of H2O2 . Catalase blocks the Ca+2+ response indicating that it is a function of H2O2
· Phamaceutical and genetic (siRNA) inhibition of TRPM2 also inhibited the Ca response to honey.
· Pharmacuetical inhibition of Orail also diminished the Ca response to honey
· Inhibition of both types of channels completely abrogated the Ca response
· Examination of aquaporin expression found that AQP3 was upregulated by honey. siRNA for AQP3 could block the Ca increase elicited by honey
Comments
· This study would be strengthened by the authors directly demonstrating H2O2 production by honey rather than relying on catalase. It is possible that H2O2 may be indirectly acting. Also, since the observed effects were dependent on the type of honey- does this correlate with H2O2 production?
· Fig 5a: Individual lanes are not identified in the immunoblot (C is probably control and M is probably honey)
· The authors should note that the keratinocyte scratch assay may have little or nothing to do with in vivo wound healing. Was “wound closure” a function of increased proliferation or migration?
Conclusion: This is a nice study with the results largely supporting the questions asked. This reviewer is a little troubled by the equating of the scratch assay with wound healing and it would be nice to see direct measurement of extracellular H2O2 . The English writing could also be improved but at present does not significantly detract from the study.

Author Response
This study would be strengthened by the authors directly demonstrating H2O2 production by honey rather than relying on catalase. It is possible that H2O2 may be indirectly acting. Also, since the observed effects were dependent on the type of honey- does this correlate with H2O2 production?
We have demonstrated by using the xylenol orange assay the H2O2 production. In Fig 3A, we evaluated the dose-dependent production of H2O2 induced by the different honeys (i.e. acacia, buckwheat and manuka). We also explored ROS production by recording the fluorescence of DHR-123 loaded HaCaT (Fig. 3B).
Fig 5a: Individual lanes are not identified in the immunoblot (C is probably control and M is probably honey)
We have corrected this mistake. In the figure legend, we have inserted that C is control and M is manuka honey exposure.
The authors should note that the keratinocyte scratch assay may have little or nothing to do with in vivo wound healing. Was “wound closure” a function of increased proliferation or migration?
Over the last decades, ethno-pharmacological studies have included in vitro assays as a replacement for experiments using tissues or whole animals (Houghton et al., J Ethnopharmacol 2005). These in vitro tests are very useful in research because of ethical reasons and of their usefulness in bioactive-guided fractionation and determination of active compounds (Ranzato et al., J Ethnopharmacol 2011).
Our previous scratch wound data and cell migration assay showed that honey improves keratinocytes wound repair capabilities.
Cell motility is a key element of tissue repair processes, and therefore, its induction could explain the ability of honey to promote the scratch wound healing of keratinocytes. Moreover, we have already performed wound healing assay in the presence of 10 μg/mL mitomycin C (MMC) for 2 h prior to the scratch assay. MMC inhibits mitosis of the cells and allowed us to distinguish between migration and proliferation. For 24-h exposure time, the main driving factor of wound closure is migration than proliferation (Martinotti et al., Molecular and Cellular Biochemistry 2017).
Conclusion: This is a nice study with the results largely supporting the questions asked. This reviewer is a little troubled by the equating of the scratch assay with wound healing and it would be nice to see direct measurement of extracellular H2O2. The English writing could also be improved but at present does not significantly detract from the study.
We have demonstrated by using the xylenol orange assay the H2O2 production. In Fig 3A, we evaluated the dose-dependent production of H2O2 induced by the different honeys (i.e. acacia, buckwheat and manuka). We also explored ROS production by recording the fluorescence of DHR-123 loaded HaCaT (Fig. 3B). We have corrected English style as well as typos present in the text.
Reviewer 2 Report
Overall impression
This study has investigated for the first time the molecular mechanisms underpinning the previously reported positive influence of ingredients of honey in wound healing both in vitro (and to a lesser extent in vivo). Using the HaCaT cell line as a skin cell in vitro model, the authors demonstrate that the H2O2 released by honey, dose-dependently induces increases in intracellular calcium concentration ([Ca2+]) most probably via entry through the AQP3 channel, which appears to drive Ca2+ influx. The demonstration that H2O2 has a defined and direct effect therefore ties in well with the antibacterial properties of honey; thus, in combination, these observations shed more light on the properties of honey. Interestingly, the H2O2-AQP3 link and wound healing also provides indirect support for the documented role of AQP3 in certain cancers, since an AQP3-orchestrated, ROS-mediated positive effect on a ‘wound-like’ response is certainly in line with this hallmark of carcinogenesis.
The study is overall well planned, the data provided appear sound and convincing, and in principle a series of novel and potentially very important observations are made. A clear strength of the study is that it provides ‘real’, molecular evidence for the reported (nearly anecdotal) positive influence of honey on wound healing. However, one main concern and weakness is the usage of the HaCaT cell line as a model.
Major points
1. Despite being widely used in the literature as a skin model, HaCaT cells do not show the same highly-proliferative, ‘basal’ cell, ‘wound-response’ phenotype that normal human epidermal keratinocytes (NHEK) demonstrate in serum-free, low Ca2+ medium. Therefore, the authors must either provide some data using NHEKs cells (for a select number of experiments to support HaCaT results), or at least make it explicitly clear (in the Abstract and Discussion) that the work was performed using this cell line.
2. The experiments described on page 6 in Section 2.3, relating to pharmacological inhibition of Ca2+ entry, are interesting and important in the context of the study. Particularly interesting is the effect observed when “HaCaT cells were pre-treated with both econazole and Pyr6…”. Yet, I do note that although this combination nicely blocked Ca2+ responses (Figure 4) and later on also evidently abrogated wound healing (Section 2.4, Figure 6), when looking carefully at the phase contrast images in Figure 6C some level of cytotoxicity is evident by the presence of classical ‘dark’ apoptotic cells (Figure 6, “eco+pyr6” image). As Ca+2 inlfux/release inhibitors can often be cytotoxic, have the authors confirmed that the combination (eco+pyr6) did not significantly affect cell viability?
3. In Section 2.4, the data in Figure 5 nicely shows that knockdown (KD) of AQP3 attenuates honey-mediated increases in [Ca2+] and by association the authors imply that AQP3 affects H2O2 influx. Yet, the link between AQP3 and H2O2 influx (implied by the title of the section – Page 8, Line 3) remains indirect; have the authors checked if AQP3 KD directly reduces H2O2 levels? The authors must either present such data or at least explicitly state this (and make a comment e.g. in the Discussion).
4. Also in Section 2.4, the experiments in Figure 6 provide evidence that blockade of H2O2 by CAT reduced wound healing. Did the authors use exogenous H2O2 treatment as a positive control / confirmatory evidence to support their hypothesis/findings?
5. In several figures the authors refer to cells and numbers of experiments carried out, e.g. on Page 3 Line 6: “20 cells from 2 exp”. Do the authors mean 20 cells from 2 independent, representative experiments (i.e. were 20 cells assessed in each experiment), or were 20 cells in total assessed and data from the 2 experiments were averaged together?
6. In Figure 5, panel F shows immunoblotting data following AQP3 KD. The figure caption states that “30-micrograms” of lysate were loaded on the gels, yet looking at the expression of beta-actin the bands look quite faint, as HaCaT certainly express a lot of actin. Can the authors confirm that 30ug of protein were indeed loaded, or was less protein analysed?
7. In the Methods, Section 4.8 (page 15) the authors provide little information on the image analysis for the scratch wound assays. However, they need to specify how they used ImageJ to analyse the images, i.e. did they measure a) wound gap or b) wound surface area. The text in the caption to Figure 6 (Page 10 Line 21) mentions “wound width”. If so, did the authors analyse several points along the wound? How many were analysed and averaged per image? Some detail would be useful. Also, how was “wound closure rate %” calculated?
8. In Figures 3, 4 and 5 the authors use lower case letters and state that “Different letters on bars indicate statistical differences…”. But what differences do ‘a’, ‘b’, ‘c’ and ‘d’ actually denote?
Minor points
1. The manuscript needs attention for grammatical issues and overall would greatly benefit from editing in terms of English language usage.
2. The authors have used a number of non-standard abbreviations throughout, such as on Page 3 Line 6 “exp” and on Page 13 Line 37 “exc”.
3. In Figure 1, the letters C and D are missing from the 2 bottom panels.
4. The schematic (Figure 7) is nice and useful but would benefit from an actual caption to explain the diagram to the reader. A caption would nicely complement the bullet-pointed summary provided in the Discussion text.
5. The sentence on Page 1 in Line 19 (Abstract) stating “…Once in the cytoplasm… (ROS)-mediated mechanism” is confusing; if H2O2 entering the cell facilitates Ca2+ entry, then isn’t this a ROS-mediated mechanism? It might be worth rephrasing this for clarity.
6. There is a plethora of errors that need attention, which include:
Page 2 Line 9, the ‘Bienert and Chaumont, 2014’ paper needs to be numbered and also in Line 16 “works” should be replaced (e.g. say “studies” instead…)
“Honey” is grammatically an uncountable noun so it does not take plural – therefore throughout the manuscript please replace “honeys” with “honey types” or “honey samples”
Page 8 Line 6, please change to “by means of qPCR data…”
Page 8 Line 15, please change to “…as shown in…”
Page 6 Line 6, please use alternative verb to “gate”
Author Response
Major points
1. Despite being widely used in the literature as a skin model, HaCaT cells do not show the same highly-proliferative, ‘basal’ cell, ‘wound-response’ phenotype that normal human epidermal keratinocytes (NHEK) demonstrate in serum-free, low Ca2+ medium. Therefore, the authors must either provide some data using NHEKs cells (for a select number of experiments to support HaCaT results), or at least make it explicitly clear (in the Abstract and Discussion) that the work was performed using this cell line.
HaCaT cells are the immortalized human keratinocyte line (Boukamp et al., 1988) extensively used to study the epidermal homeostasis and its pathophysiology.
Because HaCaT cells have a high differentiation potential in cell culture based on the expression of various epidermal differentiation markers, this cell line has been widely used as an alternative for Normal Human Epidermal Keratinocytes (NHEKs) (Grabbe et al., 1996; Lehmann, 1997).
HaCaT cell line mimics many properties of normal epidermal keratinocytes, is not invasive and can differentiate under appropriate experimental conditions (Schoop et al 1999) These cells have previously been used in wound healing studies (Ranzato et al., 2008, 2009, 2010) and display a migration index similar to that of primary human keratinocytes (Matsuura et al. 2007).
We are aware that HaCaT can possess different properties than NHEK. For our purposes, we considered sufficient to use this cellular model to highlight the involvement and production of H2O2 as a mechanism that triggers the intracellular calcium wave. To this aim, in accord to reviewer’s suggestions, we have explicit more clearly in abstract and discussion section the use of this kind of cells.
In order to define the strength and limitation of HaCaT cells and to study molecular and cellular mechanism to regulate epidermal functions under honey types exposure, we are aware that further studies using NHEK are strongly required.
2. The experiments described on page 6 in Section 2.3, relating to pharmacological inhibition of Ca2+ entry, are interesting and important in the context of the study. Particularly interesting is the effect observed when “HaCaT cells were pre-treated with both econazole and Pyr6…”. Yet, I do note that although this combination nicely blocked Ca2+ responses (Figure 4) and later on also evidently abrogated wound healing (Section 2.4, Figure 6), when looking carefully at the phase contrast images in Figure 6C some level of cytotoxicity is evident by the presence of classical ‘dark’ apoptotic cells (Figure 6, “eco+pyr6” image). As Ca+2 inlfux/release inhibitors can often be cytotoxic, have the authors confirmed that the combination (eco+pyr6) did not significantly affect cell viability?
The toxicity of Ca2+ influx/release inhibitors was negligible. The “dark” spots present in figure 6 are not apoptotic cells, but the dye inadvertently precipitated during sample staining. We changed the images, improving the quality. We apologize for the inconvenience.
3. In Section 2.4, the data in Figure 5 nicely shows that knockdown (KD) of AQP3 attenuates honey-mediated increases in [Ca2+] and by association the authors imply that AQP3 affects H2O2 influx. Yet, the link between AQP3 and H2O2 influx (implied by the title of the section – Page 8, Line 3) remains indirect; have the authors checked if AQP3 KD directly reduces H2O2 levels? The authors must either present such data or at least explicitly state this (and make a comment e.g. in the Discussion).
The referee is right. We added a novel figure, i.e. Figure 5G, to show that AQP3 KD prevented the increase in cytosolic ROS levels. We have further discussed this finding, which clearly demonstrates that AQP3 mediates H2O2 entry.
4. Also in Section 2.4, the experiments in Figure 6 provide evidence that blockade of H2O2 by CAT reduced wound healing. Did the authors use exogenous H2O2 treatment as a positive control / confirmatory evidence to support their hypothesis/findings?
Most types of honey generate H2O2, by means of enzymatic activity of gluconic oxidase, added to the nectar by bees, that converts glucose into H2O2 and gluconic acid under aerobic conditions. Some honey types does not produce H2O2 at significant levels, and several factors can affect this low production such as inactivation of glucose oxidase by exposure to heat or light or a direct inhibition by catalase originating from pollen, nectar or microorganisms.
We performed the measurements by stimulating the cells with H2O2 solution. We found that the Ca2+ response to manuka honey was mimicked by 50 µM H2O2. These new data, along with the observation that the Ca2+ response to Manuka honey was inhibited in a dose-dependent manner by CAT, an H2O2 scavenger, and with previous observations on the role of reactive oxygen species, especially H2O2, in wound healing (Loo et al., Free Radical Biology and Medicine 2011; Loo and Halliwell, Biochemical and Biophysical Research Communications 2012) strongly suggest that 50 µM H2O2 triggers this Ca2+ signal.
5. In several figures the authors refer to cells and numbers of experiments carried out, e.g. on Page 3 Line 6: “20 cells from 2 exp”. Do the authors mean 20 cells from 2 independent, representative experiments (i.e. were 20 cells assessed in each experiment), or were 20 cells in total assessed and data from the 2 experiments were averaged together?
Yes, with the statements of 20 cells from 2 exp, we mean 20 cells assessed in each experiments.
6. In Figure 5, panel F shows immunoblotting data following AQP3 KD. The figure caption states that “30-micrograms” of lysate were loaded on the gels, yet looking at the expression of beta-actin the bands look quite faint, as HaCaT certainly express a lot of actin. Can the authors confirm that 30ug of protein were indeed loaded, or was less protein analysed?
Thank you for highlighting this inaccuracy: 15 mg of protein lysate were loaded. Indeed, cells were silenced at confluence lower than 50% and used 24 h after transfection.
7. In the Methods, Section 4.8 (page 15) the authors provide little information on the image analysis for the scratch wound assays. However, they need to specify how they used ImageJ to analyse the images, i.e. did they measure a) wound gap or b) wound surface area. The text in the caption to Figure 6 (Page 10 Line 21) mentions “wound width”. If so, did the authors analyse several points along the wound? How many were analysed and averaged per image? Some detail would be useful. Also, how was “wound closure rate %” calculated?
The width of the wound space was measured at wounding and at the end of treatments, using an inverted microscope equipped with a digital camera (Leica Microsystems). Digitized pictures of wounds were analyzed using the NIH ImageJ software.
In a typical experiment, each group consisted of three different plates, i.e. a total of six wounds. Four measurements of wound width were made for each wound at randomly chosen points.
Measurements were made by a single observer unaware of the treatments. Wound closure rates were determined as the difference between wound width at 0 and 24 h.
The value of the control was set at 100 percent in order to obtain wound closure rate %.
8. In Figures 3, 4 and 5 the authors use lower case letters and state that “Different letters on bars indicate statistical differences…”. But what differences do ‘a’, ‘b’, ‘c’ and ‘d’ actually denote?
Different letters shared by the bars indicate significant differences (p<0.01) between experimental conditions as indicated in the legends.
Minor points
1. The manuscript needs attention for grammatical issues and overall would greatly benefit from editing in terms of English language usage.
We have corrected the mistakes.
2. The authors have used a number of non-standard abbreviations throughout, such as on Page 3 Line 6 “exp” and on Page 13 Line 37 “exc”.
We have corrected the mistakes.
3. In Figure 1, the letters C and D are missing from the 2 bottom panels.
We have corrected the mistakes.
4. The schematic (Figure 7) is nice and useful but would benefit from an actual caption to explain the diagram to the reader. A caption would nicely complement the bullet-pointed summary provided in the Discussion text.
We have inserted this sentence.
Data showed that honey could trigger intracellular Ca2+ changes. The levels and profile was dependent on the source of honey and the source of the Ca2+ is extracellular. Honey induced the production of micromolar levels of H2O2. Catalase blocks the Ca2+ response indicating that it is a function of H2O2 production. Pharmacological and genetic (siRNA) inhibition of TRPM2 inhibited the Ca2+ response to honey. Pharmaceutical inhibition of Orai1 also diminished the Ca2+ response to honey. Inhibition of both types of channels completely abrogated the Ca2+ response. Examination of aquaporins expression found that AQP3 was upregulated upon honey exposure. siRNA for AQP3 blocked the Ca2+ increase elicited by honey treatment.
5. The sentence on Page 1 in Line 19 (Abstract) stating “…Once in the cytoplasm… (ROS)-mediated mechanism” is confusing; if H2O2 entering the cell facilitates Ca2+ entry, then isn’t this a ROS-mediated mechanism? It might be worth rephrasing this for clarity.
We have modified this sentence
6. There is a plethora of errors that need attention, which include:
Page 2 Line 9, the ‘Bienert and Chaumont, 2014’ paper needs to be numbered and also in Line 16 “works” should be replaced (e.g. say “studies” instead…)
“Honey” is grammatically an uncountable noun so it does not take plural – therefore throughout the manuscript please replace “honeys” with “honey types” or “honey samples”
Page 8 Line 6, please change to “by means of qPCR data…”
Page 8 Line 15, please change to “…as shown in…”
Page 6 Line 6, please use alternative verb to “gate”
We have corrected the mistakes.